# Exploring Literacy and Knowledge Gaps and Disparities in Genetics and Oncogenomics Among Cancer Patients and the General Population: A Scoping Review

**DOI:** 10.3390/healthcare13020121

**Published:** 2025-01-09

**Authors:** Katerina Nikitara, Maria Luis Cardoso, Astrid Moura Vicente, Célia Maria Batalha Silva Rasga, Roberta De Angelis, Zeina Chamoun Morel, Arcangela De Nicolo, Maria Nomikou, Christina Karamanidou, Christine Kakalou

**Affiliations:** 1Hellenic Cancer Federation, Solonos Street 94, 10680 Athens, Greece; director@ellok.org; 2Instituto Nacional de Saúde Doutor Ricardo Jorge, Avenida Padre Cruz, 1649-016 Lisboa, Portugal; m.luis.cardoso@insa.min-saude.pt (M.L.C.); astrid.vicente@insa.min-saude.pt (A.M.V.); celia.rasga@insa.min-saude.pt (C.M.B.S.R.); 3BioISI-Biosystems & Integrative Sciences Institute, Faculty of Sciences, University of Lisboa, Campo Grande, C8, 1749-016 Lisboa, Portugal; 4Department of Oncology and Molecula Medicine, Istituto Superiore di Sanità, Viale Regina Elena 299, 00161 Rome, Italy; roberta.deangelis@iss.it; 5Advanced Training Office, Institut Curie, 26 rue d’Ulm, CEDEX 05, 75005 Paris, France; zeina.chamoun-morel@curie.fr; 6Center for Omics Sciences, IRCCS San Raffaele Scientific Institute, Via Olgettina 60, 20132 Milan, Italy; arcadenicolo@gmail.com; 7Institute of Oncology and Molecular Genetics, Rīga Stradiņš University, 13 Pilsoņa Street, LV-1002 Riga, Latvia; 8Centre for Research and Technology Hellas, Institute of Applied Biosciences, 57001 Thessaloniki, Greece; ckaramanidou@certh.gr (C.K.); ckakalou@certh.gr (C.K.)

**Keywords:** literacy, knowledge, cancer patients, citizens, caregivers, genetics, oncogenomics, genetic testing

## Abstract

Background: Genetic and genomic literacy is pivotal in empowering cancer patients and citizens to navigate the complexities of omics sciences, resolve misconceptions surrounding clinical research and genetic/genomic testing, and make informed decisions about their health. In a fast-evolving scenario where routine testing has become widespread in healthcare, this scoping review sought to pinpoint existing gaps in literacy and understanding among cancer patients and the general public regarding genetics and genomics. Methods: Adhering to the PRISMA framework, the review included 43 studies published between January 2018 and June 2024, which evaluated the understanding of genetics and genomics among cancer patients, caregivers, and citizens. Results: Although the selected studies had significant heterogeneity in populations and evaluation tools, our findings indicate inadequate literacy levels, with citizens displaying lower proficiency than cancer patients and caregivers. This review highlighted consistent knowledge gaps in understanding the genetic and genomic underpinnings of diseases, encompassing misconceptions about mutation types and inheritance patterns, limited awareness of available genetic testing options, and difficulties in interpreting test results. Ethical and privacy concerns and the psychological impact of genetic testing were also common, highlighting the imperative need for effective communication between healthcare providers and patients. Conclusions: Given the dynamic nature of genomic science, the review underscores the need for continuously evolving educational programs tailored to diverse populations. Our findings could guide the development of educational resources addressed explicitly to cancer patients, caregivers, and the lay public.

## 1. Introduction

Over the last twenty years, the field of oncogenomics has greatly enhanced our understanding of cancer, paving the way for new strategies in prevention, diagnosis, and treatment. Notably, the rapid development of these innovations has resulted in a significant disparity between scientific and technological advancements and cancer patients’ literacy, emphasising an immediate need to close this gap to guarantee fair access to precision oncology [1]. Genomic variants play a crucial role in the advancement of cancer, affecting various aspects such as tumour onset and responses to treatments. These genetic changes act as essential indicators, offering valuable information that aids in early diagnosis, enhances prognostic assessments, and allows for the creation of personalised treatment strategies customised for individual patients in the context of precision medicine [2]. However, the successful integration of these advancements into healthcare systems is not solely a technological challenge; it also requires robust educational efforts to empower all stakeholders. This includes not only oncologists and healthcare providers but also cancer patients, their families, and the general public [3].

Genetic and genomic literacy is inextricably linked to the concept of health literacy [4], as limited health literacy has been associated with lower genetic knowledge [5]. Both health and genomic literacy are essential for empowering cancer patients and citizens to understand omics sciences and make informed decisions about their health while limiting misconceptions about clinical research and genomic testing, fostering a more nuanced understanding and neutral perception of human diversity. Genomic literacy could also support efficient and harmonised omics data integration into healthcare [6,7]. Research has shown that lower health literacy in cancer patients is linked to reduced quality of life and care experiences, as well as increased difficulty in understanding and processing information about the disease and personal health condition [8]. Undoubtedly, health literacy holds significant importance for individuals, as it has the potential to positively impact health behaviours and outcomes while reducing health inequities. Further, improving health literacy provides a valuable advantage by mutually reinforcing other literacies, including educational, legal, financial, technological, and other forms of literacy [9]. Education in genetics and genomics is also crucial for young people, as it helps them shape their attitudes and perspectives towards personalised medicine and research and equips them with the knowledge to support their elder relatives in the field [10]. Previous studies evaluating the knowledge of citizens and patients on precision medicine, including genetic and genomics technologies, found that patients exhibit a greater familiarity with the associated terminology compared to individuals from the public [1]. Further, the extent to which a patient has been introduced to precision medicine will influence his understanding. Patients were often found to be unsure of how precision medicine tests work, and many experienced a psychological burden while awaiting and interpreting test results [1]. A scoping review on pediatric patients and their caregivers found that health literacy on genomics was low, while age was positively associated with an increased understanding of genomic concepts and perceived ability to use genomic information in decision-making [11]. Also, the most frequently expressed concerns about genetic testing are (i) the storage and management of genomic information, (ii) the privacy and confidentiality of personal data, (iii) the accessibility and affordability of testing, (iv) the apprehensions arising from associations with cloning, (v) the potential issues related with the insurance and employment discrimination, (vi) the social divisions that may arise, and (vii) the potential for family conflicts resulting from test outcomes [12].

With constant advancements in the fast-paced field of genomic science and the increasing integration of routine testing in healthcare—for diagnosis, prevention, and treatment—the need to address literacy and knowledge gaps among cancer patients, caregivers, and citizens has become increasingly apparent. This scoping review was conducted as part of the European Union-funded project “Can.Heal—Building the EU Cancer and Public Health Genomics Platform”. The primary objective of the review is to systematically examine and identify the current gaps in knowledge and literacy of cancer patients, caregivers and citizens related to genetics and genomics in the context of cancer care. Through this analysis, the study aims to inform the development of targeted, evidence-based educational programmes. These programmes will be designed to enhance the understanding of genomic science among cancer patients, carers, and the wider public, facilitating equitable access to cutting-edge cancer care and ensuring the effective implementation of precision medicine across diverse populations.

## 2. Methods

This scoping review was conducted and reported following the Preferred Reporting Items for Systematic Reviews and Meta-Analyses extension for Scoping Reviews (PRISMA-ScR) [13]. Two electronic databases (Medline and Scopus) were searched for peer-reviewed articles published between 1 January 2018 and 18 June 2024. The keywords were carefully chosen, including terms related to cancer, such as “oncology”, “neoplasms”, “malignancy, “hereditary cancer syndromes”, “ cancer predisposition”, combined with terms related to the target population like “ patient”, “citizen”, “general public”, “youth”, “ adolescent”, also combine with terms related to genomics as “gene”, “genetic”, “genome”, “genomic”, “personalised medicine”, “genetic testing”, “genetic counselling “, “next-generation sequencing”, and terms related to literacy such as “knowledge”, “understanding”, “literacy”, “awareness”. The details of the literature search strategy are outlined in Appendix A. The inclusion criteria were developed based on the PCC (P-Population, C-Concept, C-Context) framework, according to which the eligible studies should be in English, have a qualitative or quantitative study design, and explore the knowledge and understanding of genetics and oncogenomics concepts among cancer patients and/or survivors, caregivers, and families, as well as citizens of any age and sex. Given the limited number of papers addressing the topic of this review, the inclusion criteria were decided to be broad enough to capture the overall picture. Grey literature was excluded, but the references of the included studies were thoroughly searched to identify any additional relevant articles. Registration statement: This scoping review was not registered in any publicly available database prior to its conduct.

### 2.1. Study Selection

Studies retrieved through the above-mentioned electronic searches were entered into Endnote, and duplications were systematically removed. The collected studies were then imported into the Rayyan software (version 1.50) (https://www.rayyan.ai/ (accessed on 18 June 2024)) and screened based on the predefined inclusion criteria. As a first step, all studies were assessed based on their title and abstract. To test the robustness of the screening process, a pilot title/abstract screening was run by two reviewers (ΚΝ and ΜΝ) independently, covering 10% of the hits. Any discrepancies were resolved through discussion. The level of agreement between the reviewers was assessed using Cohen’s Kappa coefficient, which showed an adequate level of agreement (Cohen’s Kappa = 0.72). Due to capacity limitations, a single reviewer (ΚΝ) carried out the remaining screening. The studies selected from the title/abstract screening underwent further assessment of their full texts by a single reviewer.

### 2.2. Data Extraction and Presentation

Details about the study type and setting, the characteristics of the analysed populations, and the outcomes of interest were rigorously extracted via a standardised process. For qualitative analyses, the retrieved data included information about each study’s methodology. To test the reliability and consistency of the data extraction, two reviewers independently conducted a pilot in 10% of the studies. Once a consensus was achieved between the reviewers, one reviewer extracted data from the remaining studies. The results are presented in a tabulated format and synthesised as a narrative.

## 3. Results

### 3.1. Study Characteristics

A total of 12,993 studies were initially identified through the Medline and Scopus databases using the developed search strategy (Appendix A). After removing duplicate entries, 12,977 studies underwent title/abstract screening. Of these, 99 studies were eligible based on the predefined selection criteria and underwent full-text assessment. Sixty-one studies were excluded because of (i) limited data (n = 19), (ii) insufficient outcomes of interest (n = 38), (iii) ineligible study type (n = 3), and (iv) ineligible population (n = 1). As a result, 38 studies were included in the scoping review after successfully passing the rigorous screening process. Five studies were identified from the references of the selected studies and added to the list, thus leading to a total number of 43 eligible papers included in this scoping review. A PRISMA flow chart of the study selection steps in a comprehensive and rigorous way is presented in Figure 1.

Of the 43 studies included in the review, a total of 29 reported data on adult cancer patients and/or survivors with sample sizes ranging from 11 to 1139 participants (total N = 6358) [14,15,16,17,18,19,20,21,22,23,24,25,26,27,28,29,30,31,32,33,34,35,36,37,38,39,40,41,42]. Additionally, six studies reported data on caregivers/family members, with participants varying from 29 to 213 (total N = 617) [14,30,34,43,44,45]. Nine studies presented data on citizens with cohorts ranging from 32 individuals to a maximum of 2895 (total N = 6102) [14,46,47,48,49,50,51,52,53], and three studies presented mixed results for cancer patients and caregivers with cohorts ranging from 15 to 111 participants (total N = 152) [54,55,56].

With regards to the geographic location, twenty-five studies were conducted in the United States [16,17,18,19,20,21,23,24,26,31,34,35,37,39,40,41,43,46,47,49,51,52,53,54,56], seven studies in Australia [15,29,30,32,33,48,50], three in Canada [27,28,55], two in Republic of Korea [25,36], one in Malaysia [14], one in India [42], and one in China [45]. Only two studies were conducted in Europe, one in the Netherlands [44], and one in Ireland [22]. Finally, one study included individuals from several countries [38]. Considering the study design, 14 articles were associated with umbrella clinical studies; thus, only baseline data were retrieved and presented in this scoping review [16,18,19,20,21,23,28,29,30,32,33,34,43,51]. Sixteen studies used a cross-sectional approach [14,17,22,24,25,26,27,31,36,41,42,45,47,49,53,54], five studies had a cohort design [15,38,39,44,48], four studies applied a qualitative methodology [37,40,50,55], and four studies used a mixed methods analysis [35,46,52,56]. A detailed overview of the published studies included in the scoping review is provided in Appendix A.

### 3.2. Results on Genetic and Oncogenomic Literacy

For structured data reporting, the population was divided into four groups: (A) cancer patients and/or survivors, (B) cancer patients and caregivers (mixed results), (C) caregivers and family members, and (D) citizens. For each group, knowledge of genetics and oncogenomics concepts was categorised as follows: (i) knowledge of general genetic and genomic concepts, (ii) knowledge of genetic and genomic concepts related to general health and cancer, and (iii) knowledge of genetic and genomic testing.

#### 3.2.1. Cancer Patients and/or Survivors

Twenty-nine studies [14,15,16,17,18,19,20,21,22,23,24,25,26,27,28,29,30,31,32,33,34,35,36,37,38,39,40,41,42] assessed the overall level of knowledge of cancer patients and/or survivors on genetic and oncogenomic concepts (Table 1). In twenty-three out of the twenty-nine studies, objective knowledge was evaluated [14,15,16,17,18,19,20,21,23,24,25,26,27,28,30,31,32,34,35,36,39,41,42], and self-perceived knowledge was reported in two studies [22,40]. In the remaining four studies, either qualitative methods or unspecified methods were used. Approximately half of the studies recruited individuals who participated in clinical studies [16,18,19,20,21,23,28,29,30,32,33,34] and either were in the process of receiving or had already undergone genetic or genomic testing. Prior experience with genetic or genomic testing was also a prerequisite for inclusion in most observational studies [17,24,25,26,27,31,35,36,39]. Except for three studies [17,25,45], most participants had undergone genetic or genomic testing. Overall, the results showed that while some studies reported high literacy levels, others indicated substantial gaps.

##### Knowledge of General Genetic and Genomic Concepts

Two studies evaluated the literacy related to general genetic and genomic concepts. A high level of knowledge was found by Hamilton et al. (2019) [39] in 57 individuals with a history of cancer, scoring a mean of 0.84 (SD 0.16), ranging from 0 to 1. A slightly lower level was reported by Adams et al. (2020) [21] in 58 patients with metastatic breast cancer who scored 0.72 (score range: 0–1).

##### Knowledge of Genetic and Genomic Concepts Related to General Health and Cancer

Thirteen studies assessed the knowledge of genetic/genomic concepts related to cancer, among which eleven consistently showed an overall moderate level of understanding, and two indicated poor knowledge. Specifically, Butow et al. (2022) [15] found a mean score of 47.9% (SD 30.1%, n = 261) in a cohort of individuals with a personal history of cancer. Similarly, Pozzar et al. (2022) [24] estimated a mean score of 11.9 (SD 3.5, score range: 0–19) among 87 gynaecological/breast cancer patients. Two additional studies focusing on breast cancer patients, both conducted by Underhill-Blazey et al. [26,35], also indicated a moderate level of knowledge. In the most recent one [35], a mean genetic knowledge score of 12.3 (SD 3.4, n = 602) (score range: 0–19) was observed, while in the earlier one [26], a mean knowledge score of 10 (SD 3, n = 591) (score range: 0–16) was obtained. Likewise, participants (n = 85) in a study by Wing et al. (2021) [17] answered five out of nine questions correctly on average, while 58.6% of the cancer patients in the study by Aizuddin et al. (2021) [14] obtained a moderate or low score regarding their knowledge of cancer genetic/genomic concepts. Similar findings were reported by Dehar et al. (2022) [27] in a cohort of 113 cancer patients by Liang et al. (2018) [31] in 53 ovarian cancer patients, by Marron et al. (2019) [34] in 11 cancer patients, and by Gornick et al. (2018) [23] in a cohort of 537 newly diagnosed patients with early-stage breast cancer, where no detailed numerical data were provided. Finally, qualitative results from Frost et al. (2019) [38], reflecting 32 individuals with a history of cancer or increased cancer risk, showed a moderate level of understanding.

A low level of understanding was reported in two studies [40,42]. The survey of Pramanik et al. (2024) [42] in 84 women with breast or ovarian cancer revealed a mean knowledge score of 5.11 (SD 2.54), with a range of 0–13, amounting to 39.3  ±  19.5%. Also, the qualitative findings of Robles-Rodriguez et al. (2024) [40] showed that women with breast cancer (n = 29) were unfamiliar with precision medicine, with 42% believing that genes have little effect on health.

##### Knowledge of Genetic and Genomic Testing

Twenty studies assessed comprehension of genetic and genomic testing by cancer patients or survivors. Of these, five studies reported a high level of knowledge [20,23,27,28,31], seven documented a moderate level [14,16,17,19,25,30,36], and the remaining eight studies identified either a low or moderate [18,22,32] or a low level of understanding [29,33,37,40,41].

Roberts et al. (2019) [20] revealed a high level of understanding of genetic/genomic testing in 217 patients with treatment-resistant, metastatic cancer, achieving an average score of 5.3 (SD 0.99) out of 6. Likewise, McCuaig et al. (2021) [28] reported a mean knowledge score of 7.8 (SD 2.1) (score range: 0–10) among 120 patients with breast and ovarian cancer, while Dehar et al. (2022) [27] observed that over 80.5% of the 113 adult cancer patients in their study comprehended the purposes of genetic testing. Similarly, a high level of knowledge concerning the benefits and purposes of genetic testing was reported by Gornick et al. (2018) [23] among 537 newly diagnosed patients with early-stage breast cancer and by Liang et al. (2018) [31] in 53 patients diagnosed with epithelial ovarian cancer.

A moderate level of knowledge of genetic and genomic testing was found by Makhnoon et al. (2021) [16], obtaining a mean score of 10.6 (score range: 0–22) from 18 colon cancer patients, whereas Shin et al. (2021) found a score of 6 (score range: 0–11) among 103 ovarian cancer patients. Similarly, Park et al. (2022) [36] estimated a mean knowledge score of 66.9% (SD 21.7%) from a cohort of 700 BRCA1/2 mutation-negative breast cancer patients, while Napier et al. (2022) [30] indicated a relative score of 45% (SD 25%) in a population of 348 patients diagnosed with a likely hereditary form of cancer. Likewise, in the study by Wing et al. (2021) [17], the average number of correct answers given by 85 cancer patients was 10 (SD 5) out of 19, while Roth et al. (2021) [19] reported a moderate proportion of correct answers on genetic testing among 207 participants with advanced non-small-cell lung cancer. Finally, Aizuddin et al. (2021) [14] found a moderate level of knowledge in more than half of the 86 participating cancer patients.

A low to moderate level of knowledge was observed in three studies. In the research by Anderson et al. (2021) [18], which included 1139 cancer patients, a percentage of 48% provided correct answers with a standard deviation of 31%, evidencing a notable heterogeneity in the level of knowledge. In a study by Mullally et al. (2021) [22], 58% (n = 49) of cancer patients had little or no knowledge about genetic testing. Additionally, according to Davies et al. (2020) [32], adult patients with confirmed advanced or metastatic solid cancers (n = 777) displayed poor to moderate knowledge about molecular tumour profiling, scoring an average of 43% (SD 20%) incorrect responses. Finally, a low level of knowledge regarding genetic and genomic testing was identified in five studies. The survey by Wang et al. (2023) [41] resulted in an average genetic testing knowledge score of 1.90 (SD = 1.48; range 0–7), suggesting a low level of genetic knowledge, while the qualitative studies by Bartley et al. (2020) [29], Best et al. (2019) [33], Gómez-Trillos et al. (2020) [37], and Robles-Rodriguez et al. (2024) [40] showed that although participants understood the concept of genetic testing, they had difficulties relating genomic testing to personalised medicine.

#### 3.2.2. Cancer Patients and Caregivers (Mixed Results)

Three studies (Table 2) presented mixed results for cancer patients and caregivers [54,55,56]. Oberg et al. (2018) [54], who analysed 111 parents of pediatric cancer patients and young adult cancer survivors, found (i) a mean score of 4.11 (SD 1.41) (score range: 1–7) for general genetic concepts, (ii) a mean score of 8.07 (SD 2.37) (score range: 1–12) for genetic concepts related to general health and cancer, and (iii) a mean score of 6.18 (SD 4.44) (score range: 0–16) for sequencing-related concepts. The study by Hill et al. (2018) [55], focusing on 15 parents of children with retinoblastoma and adult retinoblastoma survivors, showed that although the participants generally understood that retinoblastoma is a genetic disease, concepts related to retinoblastoma genetics were often misunderstood. Finally, Stallings et al. (2023) [56], who performed a mixed-methods analysis, concluded that the 26 enrolled individuals with personal cancer experience (patients or caregivers) were not familiar with precision medicine concepts.

#### 3.2.3. Caregivers and Family Members

Six studies [14,30,34,43,44,45] (Table 3) recruited caregivers and family members, hereinafter mentioned as caregivers. The identified studies evaluated two of the three domains of knowledge considered in this review.

##### Knowledge of Genetic and Genomic Concepts Related to General Health and Cancer

A moderate to high level of knowledge was identified in two studies. According to Johnson et al. (2019) [43], a median percentage of 77.8% correct answers were obtained from 158 parents of children with cancer. Marron et al. (2019) [34] reported a high level of genetic knowledge in twenty-four participants and a low level in eight. A lower level of knowledge was demonstrated by Aizuddin et al. (2019) [14], who found that only 39.4% of 57 caregivers had adequate knowledge, and Bon et al. (2022) [44], who highlighted difficulties in understanding genetic concepts by 29 parents of cancer patients.

##### Knowledge of Genetic and Genomic Testing

Regarding genetic and genomic testing, one study [45] reported a high level of knowledge, while the remaining two [14,30] agreed on a low to moderate level. Specifically, a high level of knowledge was demonstrated by Xiao et al. (2020) [45], with a median total knowledge score of 5 (range: 2–7) obtained from 126 parents of children with retinoblastoma. In contrast, in the study by Aizuddin et al. (2019) [14] involving 57 caregivers, only 33.8% scored high based on the tool measuring genetics and genomics knowledge, while in the study by Napier et al. (2022) [30], the median score of 213 caregivers was 43% (SD 25%), implying a low to moderate level of knowledge.

#### 3.2.4. Citizens

Nine studies [14,46,47,48,49,50,51,52,53] (Table 4) focused on citizens. One presented qualitative data [50], and the remaining eight reported quantitative findings. Objective knowledge was evaluated in six out of the eight quantitative studies [14,46,47,48,49,51], while subjective knowledge was measured only in two studies [52,53].

##### Knowledge of General Genetic and Genomic Concepts

The level of knowledge of general genetic and genomic concepts was estimated and found to be moderate in two studies [46,50]. In the survey conducted by Puryear et al. (2017) [46], involving 97 primary care adult patients, the mean knowledge score was 6.6 (SD 3.6) (score range: 5–12). Also, in the qualitative study by Metcalfe et al. (2018) [50], which included 56 non-expert members of the public, varying levels of awareness and understanding of genetic concepts were observed, overall classified as fairly limited.

##### Knowledge of Genetic and Genomic Concepts Related to General Health and Cancer

Two studies consistently reported low knowledge of cancer-related genetic and genomic concepts in citizens. According to Aizuddin et al. (2021) [14], only 19.2% of the 32 participating community members had an adequate level of knowledge. Further, findings by Guo et al. (2022) [47], based on the responses of 677 adult women from low-income areas, revealed an overall low knowledge of genes and cancer risk.

##### Knowledge of Genetic and Genomic Testing

Overall, seven studies assessed citizens’ level of knowledge about genetic and/or genomic testing. Four studies specifically addressed cancer [14,48,52,53], while the remaining three did not provide specific references to cancer [49,50,51]. A moderate to high level of knowledge was identified in four studies [48,51,52,53]. Horrow et al. (2019) [51] reported a mean knowledge score of 8.1 (SD 2.5) (score range: 0–11) in a cohort of 2895 adults. Similarly, Alvord et al. (2020) [52] found a mean level of knowledge of 1.90 (SD 0.7) (score range: 0–4) among 203 participants, and Fogleman et al. (2019) [53] suggested that more than two-thirds of the 114 participants in the survey (69.0%) were aware of genetic screening modalities for cancer. Finally, among the 150 general practice patients included in the study by Saya et al. (2022) [48], 73% (95% CI: 65–80%) had a knowledge score of 8 about genetic testing (score range: 0–11). The remaining three studies reported lower levels of knowledge of genetic testing. According to Krakow et al. (2018) [49], only 57.08% of 1878 adults were aware of genetic health tests. Similarly, in the study by Aizuddin et al. (2021) [14], only 19.2% of the community members had an adequate level of knowledge about genetic and genomic testing. A low level of knowledge was also reported in the qualitative study by Metcalfe et al. (2018) [50], with very few participants having heard about “direct-to-consumer” testing and only 7 out of 56 reporting having undergone genetic testing.

### 3.3. Factors Influencing the Level of Genetics/Genomics Knowledge

Fourteen out of forty-three studies [15,16,21,23,24,25,29,30,32,34,35,45,48,49] included in the present scoping review explored the relationship between the level of genetics/genomics knowledge and various socio-demographic factors. Education was the most frequently examined factor, displaying a positive association with knowledge levels in nine studies [15,16,23,30,32,34,35,45,48]. Conversely, age exhibited a negative correlation with knowledge levels in five studies [16,23,24,35,49]. Also, race/ethnicity played a role, whereby being white/non-Hispanic and not belonging to a minority group was linked to higher knowledge levels in five studies [21,23,24,41,49]. A higher income demonstrated a positive relationship with knowledge levels in three studies [21,24,49], as did, not surprisingly, having a medical background [15,29,30]. A familial history of cancer in a first or second-degree relative was associated with greater knowledge in two studies [25,30], along with prior personal or familial experience with genetic testing [29,35] and the use of English as the primary language at home [32,48]. Furthermore, individuals with higher health literacy exhibited greater knowledge in one study [15]. Women displayed a significantly higher level of knowledge compared to men in one study [48], as did individuals with a personal history of cancer [24].

## 4. Discussion

Drawing on quantitative and qualitative data from 43 studies across seven countries, this scoping review evaluates the literacy levels and understanding of genetics and genomics among cancer patients, caregivers, and the public, focusing on implications for cancer prevention, diagnosis, and treatment. Despite the heterogeneity in the samples and methods of the included studies, our findings uncover inadequate knowledge levels among all studied populations, with lower levels in citizens compared to cancer patients and caregivers. This difference is likely ascribable to the limited exposure of the former category to these concepts and practices. Most cancer patients and caregivers in the selected studies of this review had been recruited in a clinical trial context, with genetic or genomic testing being part of the protocol. Hence, participants received basic genetic information as a prerequisite for informed consent. Consistent findings were reported in the systematic review by Botham et al. (2021), revealing that patients participating in clinical trials comprehended personalised medicine concepts (and terms) better than those undergoing testing with the only purpose of informing their treatment [1]. Providing educational support before enrollment in cancer clinical trials improves the probability of participants’ acceptance [57]. This could explain the higher levels of knowledge shown by patients participating in clinical trials [58]. Nevertheless, though most participants had received genetic or genomic testing, there was still a significant lack of knowledge and misconceptions regarding interpreting the results. This stresses the importance of effective communication between cancer patients and their healthcare providers and the constant need for tailored and up-to-date education of all stakeholders [59].

The findings of this review highlight several factors influencing cancer patients’ understanding of genomic concepts, with education emerging as the most frequently examined and positively correlated factor. Higher knowledge levels were consistently associated with individuals who had access to educational resources or had a familial history of cancer, prior personal or familial experience with genetic testing, and higher health literacy, underscoring the importance of targeted educational initiatives. However, the complexity of genomic concepts, combined with limited educational resources, remains a significant barrier to widespread understanding. These findings suggest that addressing the educational gaps, particularly by simplifying complex genomic concepts and ensuring equitable access to resources, is crucial for improving understanding across diverse patient populations.

The knowledge gaps in this scoping review align with those reported in previously published reviews. The primary challenges cancer patients, caregivers, and the public face include comprehending genetics’ role in cancer and other genetic diseases, distinguishing between germline and somatic sequence variants, understanding the inheritance patterns of specific cancer-related genetic mutations, and assessing familial cancer risk. Complexities in interpreting genetic test results also emerge, including the implications of specific genetic variants [1,58]. Ethical and privacy concerns are repeatedly raised since patients and citizens seem unfamiliar with the laws regulating the use of applied genetic information and preventing discrimination regarding eligibility for life insurance, disability insurance, and long-term care insurance. Finally, misconceptions about the practical processes of sample extraction, storage, and data protection have also been observed [52].

Another relevant finding of our work is that genetic literacy can be influenced by several factors, especially education and age. Education has been generally linked with higher health literacy, suggesting that individuals with higher education tend to more appropriately seek, interpret, understand, and apply health information to make informed decisions about their health and well-being [60]. In line with what was previously reported [11], younger individuals showed better genetic knowledge than older participants, possibly owing to their increased potential exposure to information through educational curricula and the Internet [61]. Lastly, another significant factor frequently related to higher genetic literacy is personal or family history of cancer, as individuals might have a greater interest in learning about the genetic aspects of the disease. Personal experience can motivate individuals to seek information, engage in discussions, and develop a better understanding of genetic factors contributing to cancer development [62]. It is important to note that genetic literacy is a complex and evolving field, and various other factors can influence an individual’s understanding of genetic information.

Based on our results, several strategies can be developed to improve literacy across these groups. For instance, targeted educational initiatives could focus on simplifying complex genomic concepts and using accessible language tailored to the literacy levels of different populations. Incorporating interactive and personalised learning approaches, such as digital platforms or community-based workshops, could also increase engagement, particularly for groups with limited access to resources. Additionally, integrating these educational efforts into routine clinical care and leveraging the support of healthcare providers, including genetic counsellors, could ensure that the information is both accurate and contextually relevant. By fostering a more knowledgeable population through these tailored approaches, individuals will be better empowered to make informed decisions about their health and participate actively in their cancer care.

### Strengths and Limitations

This scoping review provides the most up-to-date evidence reflecting the knowledge of genetic and genomic concepts by cancer patients, caregivers, and citizens, synthesising the existing body of quantitative and qualitative data. We carried out a comprehensive literature search, adopting the best available standards to select and analyse the collected evidence—an approach that strengthens the methodology of this scoping review. However, the current scoping review was subject to some limitations due to the breadth of studies that were eligible for analysis. As mentioned earlier, most of the included studies were conducted in the United States, while only two were carried out in Europe, limiting the generalisability of our findings. In this regard, another factor that should be considered is the participation of a significant number of patients in clinical trials for which they were offered genetic or genomic testing, meaning that they had already been exposed to essential information for these concepts to be able to provide informed consent. Also, included studies predominantly addressed cancers such as breast and ovarian cancer. While cancers affecting both sexes are equally important, the representation of such data in the literature we reviewed was comparatively limited. Apart from that, while cultural and societal factors play a critical role in shaping genetic literacy and perceptions of genomic testing, our review explores these aspects only partially due to insufficient evidence in the original studies and under-representation of minorities and underserved populations. Furthermore, cancer patients, caregivers, and citizens are grouped according to the definitions provided in the original studies, limiting their comparability. Significant heterogeneity was noted regarding the type and the stage of cancer among cancer patients, whereas the cohorts of caregivers mainly encompassed parents acting as legal representatives of their underage children. Lastly, significant heterogeneity was identified in the knowledge assessment methods and tools, including variations in the type of acquired knowledge, difficulty levels, complexity, and the number of questions included, while the included studies were not assessed for their methodological quality.

## 5. Conclusions

In conclusion, the findings from this scoping review highlight variable levels of genetic and oncogenomic literacy among cancer patients, caregivers, and the public, revealing significant gaps that, if addressed, could enhance patient engagement and health outcomes.

## Figures and Tables

**Figure 1 healthcare-13-00121-f001:**
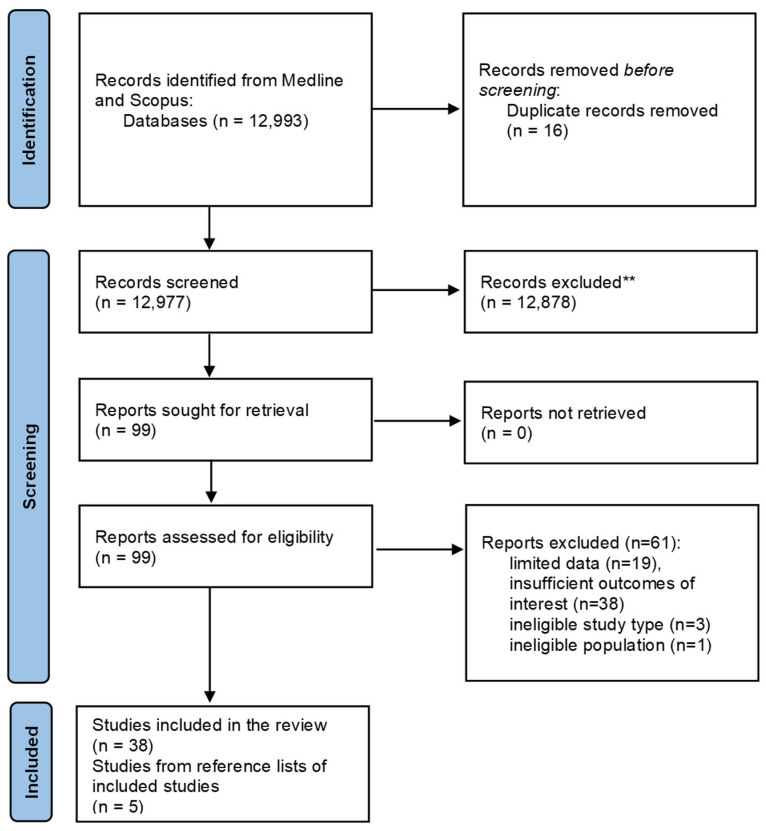
PRISMA flowchart of study selection. ** Based on title/abstract screening.

**Table 1 healthcare-13-00121-t001:** Results for cancer patients’ and/or survivors’ knowledge of genetics and oncogenomics. Colour explanation: green: high literacy level/yellow: moderate literacy level/orange: low literacy level. N/A: Not Applicable.

Author, Year	Knowledge of General Genetic/Genomic Concepts	Knowledge of Genetic/Genomic Concepts Related to Cancer	Knowledge of Genetic/Genomic Testing
Aizuddin et al., 2021 [14]	N/A	High score: 41.4% (Score for high: 6–10)	50.6% scored high (Score for high: 6–10)
Butow et al., 2022 [15]	N/A	Mean score: 47.9% (SD = 30.1%, n = 261)	N/A
Makhnoon et al., 2021 [16]	N/A	N/A	Average score: 48.2% (10.6 of 22 total possible points)
Wing et al., 2021 [17]	N/A	Average correct questions: 5 (SD = 2) out of 9	Average correct questions: 10 (SD = 5) out of 19
Anderson et al., 2021 [18]	N/A	N/A	48% correct answers (SD 31%)
Roth et al., 2021 [19]	N/A	N/A	The proportion providing correct answers to these questions was moderate
Roberts et al., 2019 [20]	N/A	N/A	Average score of 5.3 (SD = 0.99) out of 6 items (88% correct answers)
Adams et al., 2020 [21]	Mean score of 0.72 (range: 0–1)	N/A	N/A
Mullally et al., 2021 [22]	N/A	N/A	58% (n= 49) declared little or no knowledge
Gornick et al., 2018 [23]	N/A	Low level: 29.8% correct answers	High level: 72.49–89.20% correct for each question
Pozzar et al., 2022 [24]	N/A	Mean score: 11.9 (SD = 3.5) out of 19	N/A
Shin et al., 2021 [25]	N/A	N/A	Mean score: 6 (range of 0–11)
Underhill-Blazey et al., 2021 [35]	N/A	Mean score: 12.3 (SD 3.4) out of 19/on average, participants answered 63% of questions correctly	N/A
Park et al., 2022 [36]	N/A	N/A	Mean score: 66.9 (SD 21.7) (range 0–100)
Marron et al., 2019 [34]	N/A	4 participants had high genetic knowledge, and 5 had low	N/A
Underhill-Blazey et al., 2019 [26]	N/A	Mean knowledge score:10 (SD 3) (range 0–16)	N/A
Dehar et al., 2022 [27]	N/A	Moderate	Moderate to high
McCuaig et al., 2021 [28]	N/A	N/A	Mean score: 7.8 (SD 2.1) (range 0–10)
Bartley et al., 2020 [29]	N/A	N/A	85% of participants acknowledged that they did not fully understand or were uncertain about what genome sequencing is
Napier et al., 2022 [30]	N/A	N/A	Mean score: 45% (SD 25)
Liang et al., 2018 [31]	N/A	Moderate	High level of knowledge
Davies et al., 2020 [32]	N/A	N/A	Overall, poor to moderate knowledge with an average correct response score of 43% (SD 20%)
Best et al., 2019 [33]	N/A	N/A	Participants’ understanding was generally poor
Gómez-Trillos et al., 2020 [37]	N/A	N/A	Low level of knowledge of genetic services
Frost et al., 2019 [38]	N/A	Μoderate level of familiarity with/interest in genetic	N/A
Hamilton et al., 2019 [39]	Mean score: 0.84 (SD 0.16) (range 0 to 1)	N/A	N/A
Robles-Rodriguez et al., 2024 [40]	N/A	Confused about precision medicine, with 42% believing that genes have little effect on health	Participants understood the concept of genetic testing, but they had difficulties relating genomic testing to personalised medicine
Pramanik et al., 2024 [42]	N/A	Mean score: 5.11 (SD 2.54) (range: 0–13)	N/A
Wang et al., 2023 [41]	N/A	N/A	Mean score: 1.90 (SD = 1.48) (range 0–7)

**Table 2 healthcare-13-00121-t002:** Results for cancer patients and caregivers’ knowledge of genetics and oncogenomics. Colour explanation: green: high literacy level/yellow: moderate literacy level/orange: low literacy level. N/A: Not Applicable.

Author, Year	Knowledge of General Genetic/Genomic Concepts	Knowledge of Genetic/Genomic Concepts Related to Cancer	Knowledge of Genetic/Genomic Testing
Oberg et al., 2018 [54]	Mean score: 4.11 (SD 1.41) (range 0–7)	Mean score: 8.0 (SD 2.37) (range 0–12)	Mean score: 6.0 (SD 4.44) (range 0–16)
Hill et al., 2018 [55]	N/A	Variable (often limited) knowledge of retinoblastoma genetics	N/A
Stallings et al., 2023 [56]	N/A	Low familiarity ratings for precision medicine-related terms	N/A

**Table 3 healthcare-13-00121-t003:** Results for caregivers’ knowledge of genetics and oncogenomics. Colour explanation: green: high literacy level/yellow: moderate literacy level/orange: low literacy level. N/A: Not Applicable.

Author, Year	Knowledge of Genetic/Genomic Concepts Related to Cancer	Knowledge of Genetic/Genomic Testing
Aizuddin et al., 2021 [14]	High: 39.4% (Score for high: 6–10)	High: 33.8% (Score for high: 6–10)
Johnson et al., 2019 [43]	Median percentage of total correct answers: 77.8%/54% of the participants had 75–100% correct answers	N/A
Bon et al., 2022 [44]	Parents faced difficulties grasping genetic concepts	N/A
Xiao et al., 2020 [45]	N/A	Median total score: 5 (range: −2–7)/Less than one-third of parents (n = 37, 29.4%) correctly answered all 7 questions
Marron et al., 2019 [34]	24 participants had high genetic knowledge and 8 had low	N/A
Napier et al., 2022 [30]	N/A	Mean knowledge score: 43% (25%)

**Table 4 healthcare-13-00121-t004:** Results for citizens’ knowledge of genetics and oncogenomics. Colour explanation: green: high literacy level/yellow: moderate literacy level/orange: low literacy level. N/A: Not Applicable.

Author, Year	Knowledge of General Genetic/Genomic Concepts	Knowledge of Genetic/Genomic Concepts Related to Cancer	Knowledge of Genetic/Genomic Testing
Aizuddin et al., 2021 [14]	N/A	19.2% Score for high knowledge	15.6% Score for high knowledge
Puryear et al., 2017 [46]	Mean score: 6.6 ± 3.6/12 (Score range −5 to 12)	N/A	N/A
Guo et al., 2022 [47]	N/A	Low	N/A
Saya et al., 2022 [48]	N/A	N/A	73% (95% CI: 65–80%) had adequate knowledge
Krakow et al., 2018 [49]	N/A	N/A	57.08% were aware of genetic health tests
Metcalfe et al., 2018 [50]	Low to moderate	N/A	Low
Horrow et al., 2019 [51]	N/A	N/A	Mean score: 8.1 (2.5), 0.0–11.0
Alvord et al., 2020 [52]	N/A	N/A	Mean score: 1.90 (SD = 0.7), 0–4
Fogleman et al., 2019 [53]	N/A	N/A	69.0% were aware of genetic screening modalities for cancer

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
