# Peer review of "Exploring Literacy and Knowledge Gaps and Disparities in Genetics and Oncogenomics Among Cancer Patients and the General Population: A Scoping Review"

_healthcare, 2025, doi:10.3390/healthcare13020121_

Round 1

Reviewer 1 Report

Comments and Suggestions for Authors

Dear Author/s,

The article cannot be accepted in its current state and requires Major revisions before being considered for publication. Kindly see the points below for the suggestions:

Abstract:

·         Restructure this sentence [This scoping review aimed to identify current literacy and knowledge gaps among cancer patients and citizens on matters related to genetics and genomics].

·         Would be better if you could alter these words […the lay public].

Introduction:

·         The opening sentence is informative but generic. It could be more engaging to capture the reader’s attention. Start with an impactful statement or statistic about the gap between oncogenomics advancements and cancer patients' literacy.

·         Change the heading of this section from ‘Backgrounds’ to ‘Introduction’

·         The font of this line does not match the rest of the text in the article. Please correct this. [It is crucial for empowering cancer patients, caregivers, and citizens to understand omics sciences].

·         The introduction references low literacy and knowledge gaps but provides no specific data or examples to substantiate these claims. Include a brief reference to relevant statistics or studies

·         It would be more suitable to include a few lines on the content of genomic literacy, like what it contains and what areas it addresses [Genomic literacy could also support an efficient and harmonized integration…]

·         The aim of the review needs to be more prominent. It currently feels buried within the paragraph. Move the aim closer to the end of the introduction and refine it for clarity

·         Overall, the introduction is too short in length. Please increase the word count to 700 words.

Methods:

·         While Tables S1 and S2 are referenced for the search strategy, a summary of the keywords and filters applied would enhance understanding. Please mention if the search was limited by language (e.g., English) and whether grey literature was excluded.

·         The inclusion criteria based on the PCC framework are clear but could benefit from a brief explanation of the rationale for these criteria. For example, why were qualitative and quantitative designs included, and how do they align with the scoping review's objectives?

·         You mentioned that a single reviewer took part in full-text screening, which may raise concerns about potential bias. While the pilot screening improves reliability, adding a justification for why one reviewer conducted the remaining screening would be helpful.

·         Please provide names of specific data extraction and management tools or software.

·         The Kappa coefficient is presented but lacks interpretation. Indicating what 0.72 means (e.g., 'substantial agreement') would add clarity.

Results:

·         Break the "Study Characteristics" section into Population and Geographic Distribution subsections to improve readability.

·         Include a brief explanation of the PRISMA flowchart beyond mentioning it in Figure 1, such as how it reflects the rigorous selection process.

·         Please consider summarizing the variability in knowledge levels in the section on genetic literacy (such as ‘While some studies report high literacy levels, others indicate substantial gaps, particularly among specific subgroups like breast cancer patients’)

·         Please include a new table or chart summarizing study characteristics like sample sizes, populations, and methodologies for better visual representation.

·         Clearly group studies into high, moderate, and low knowledge categories for better readability.

Discussion:

·         Briefly address reasons for low understanding, such as the complexity of genomic concepts or limited educational resources.

·         Please restructure this sentence [This difference is likely ascribable to the limited exposure of the former category to these concepts and practices….]

Conclusion:

·         It would be more suitable here to provide some strategies that can be developed based on the context of this article [development of targeted educational initiatives…].

Additional:

·         The in-text citations do not follow the journal guidelines. Please correct this throughout the article.

Reviewer 2 Report

Comments and Suggestions for Authors

Recommendation : Major Revision

It’s unclear what is the role of frontline workers/service providers and care givers in hospital in providing awareness to patients or family members in the manuscript as they play an important role in making the family or patient understand the importance of genetics/oncogenomic.

It’s unclear how many cancer patients have diseased due to the lack of awareness or does the awareness has any influence on mortality rates/survival rates.

The MS missing the section of how to improve awareness in patients or family members on oncogenomic.

Majority of the data is relevant to female cancers. How about cancers that relevant to both sex. It’s unclear what is the take home message from this study how understanding oncogenomic helps in better patient care.

Cancer patients, caregivers, and citizens are grouped and compared, but the criteria for these groupings are not adequately justified. For example, caregivers and family members are treated as a homogenous group, despite their potentially diverse roles and experiences.

The grouping of "citizens" is overly broad and lacks specificity, resulting in vague interpretations of their knowledge levels.

Studies with mixed results are summarized without delving deeply into why discrepancies exist (e.g., Oberg et al. vs. Hill et al.). This weakens the analysis and reduces the impact of the discussion.

Minimal effort is made to reconcile or explain inconsistencies between studies with conflicting findings (e.g., moderate vs. low knowledge levels).

The results heavily rely on studies conducted in the United States, with only two studies from Europe, limiting the global applicability of the findings. This issue is acknowledged but not addressed adequately in terms of potential mitigation strategies.

While knowledge gaps are frequently highlighted, the manuscript provides limited actionable insights or recommendations for addressing these gaps, particularly for the broader population of "citizens."

The manuscript relies heavily on numerical scores (e.g., mean scores, median percentages), which are not always contextualized or explained in detail. This makes it harder for readers to grasp the practical implications of the findings.

Although ethical concerns and misconceptions are briefly mentioned, there is little exploration of how cultural or societal factors may influence genetic literacy or perceptions of genomic testing.

The discussion of socio-demographic factors like education, age, and income is superficial, failing to explore the broader implications or causal relationships.

The manuscript lacks a critical evaluation of the potential biases inherent in the selected studies, such as small sample sizes (e.g., Hill et al., Marron et al.) or recruitment from clinical trial settings.

Minorities and Underserved Populations, While their underrepresentation is noted, the background does not propose how this issue might be addressed in the study.

While ethical and psychological concerns are mentioned, their connection to literacy and knowledge gaps is not elaborated. It could be clarified how these concerns directly relate to or exacerbate the identified gaps.

While cancer patients are mentioned, the emphasis often shifts to general genomic literacy, which may dilute the focus on the target population.

Statements like "genomic literacy is reportedly low" (7) lack specifics on how this was measured or defined.

The conclusion emphasizes the need for educational programs but does not specify the types of resources or strategies that might be effective. Including a hint of actionable recommendations could make the findings more impactful.

Round 2

Reviewer 2 Report

Comments and Suggestions for Authors

Accept in current form

Author Response

Dear Academic Editor,

We sincerely thank you for your insightful comments and suggestions, which have contributed significantly to improving the quality of our manuscript. We have carefully addressed all the comments as follows:

Comment 1: Write reistration statemnt as  in ethic question 5
Response 1: The statement has been added

Comment 2: Revise the conclusion: the author should not make suggestions in a conclusion like this...
Response 2: We have amended the text accordingly. Please see in track changes.

Comment 3: The reviewer 2 has  provided many comments related to limitations of the review, the PI should add this in the limitations as far as possible. such as comments 14, 15.
Response 3: We already have limitation statements for comments 4,7,8,9 and 12. We added statements for comments 5,14,15.